

# A novel genus and cryptic species harboured within the monotypic freshwater crayfish genus *Tenuibranchiurus* Riek, 1951 (Decapoda: Parastacidae)

Kathryn L. Dawkins[1], James M. Furse[2,3], Clyde H. Wild[2] and Jane M. Hughes[4]

[1] Australian Rivers Institute, Griffith University, Gold Coast, Queensland, Australia
[2] Environmental Futures Research Institute, Griffith University, Gold Coast, Queensland, Australia
[3] Miyazaki International College, Miyazaki, Japan
[4] Australian Rivers Institute, Griffith University, Nathan, Queensland, Australia

## ABSTRACT

Identifying species groups is an important yet difficult task, with there being no single accepted definition as to what constitutes a species, nor a set of criteria by which they should be delineated. Employing the General Lineage Concept somewhat circumvents these issues, as this concept allows multiple concordant lines of evidence to be used as support for species delimitation, where a species is defined as any independently evolving lineage. Genetically diverse groups have previously been identified within the monotypic parastacid genus *Tenuibranchiurus* Riek, 1951, but no further investigation of this diversity has previously been undertaken. Analysis of two mitochondrial DNA gene regions has previously identified two highly divergent groups within this taxon, representing populations from Queensland (Qld) and New South Wales (NSW), respectively. Additional testing within this study of both mitochondrial and nuclear DNA through species discovery analyses identified genetically diverse groups within these regions, which were further supported by lineage validation methods. The degree of genetic differentiation between Qld and NSW populations supports the recognition of two genera; with Qld retaining the original genus name *Tenuibranchiurus*, and NSW designated as *Gen. nov.* until a formal description is completed. Concordance between the species discovery and lineage validation methods supports the presence of six species within *Tenuibranchiurus* and two within *Gen. nov.* The recognition of additional species removes the monotypy of the genus, and the methods used can improve species identification within groups of organisms with taxonomic problems and cryptic diversity.

# INTRODUCTION

Species are the fundamental unit of biodiversity, yet there has always been disagreement about criteria by which they should be recognised and the methods by which they should

Corresponding author
Kathryn L. Dawkins,
kathryn.dawkins1@gmail.com

be delineated, with no general consensus reached thus far. The lack of one clearly accepted definition of a ''species'' creates obvious limitations, as what one person regards as a species may not be regarded as being so by another person, which is often further exacerbated by differences of opinion between fields of study. Employing the General Lineage Concept (GLC; *De Queiroz, 1998*), where a species is defined as a metapopulation lineage evolving separately from other lineages, somewhat unites the various species concepts by allowing any evidence of lineage separation (and thus any property emphasised by the alternative concepts) to be used as evidence for species delimitation (*De Queiroz, 2007*). Not only does this concept allow multiple lines of evidence to be used, but it also allows the evolutionary processes that have caused divergence between lineages to be examined.

Identifying species within freshwater crayfish has traditionally been undertaken through morphological examination. However, due to the tendency of crustaceans to contain both morphologically plastic or cryptic forms (e.g., *Austin & Knott, 1996*; *Breinholt, Porter & Crandall, 2012*; *Murphy & Austin, 2003*; *Silva et al., 2010*), there has been an increasing shift towards the use of molecular methods to identify cryptic diversity (*Bentley, Schmidt & Hughes, 2010*; *Dawkins et al., 2010*; *Hansen et al., 2001*; *Mathews et al., 2008*; *Schultz et al., 2007*; *Sinclair et al., 2011*). With the use of molecular techniques comes the potential for signatures of population-level and species-level histories to become confounded, however (*Edwards, 2008*). This can occur when gene trees constructed from a single locus differ from the true genealogical history of a species (*Hey & Machado, 2003*; *Sunnucks, 2000*), although this problem can potentially be overcome by estimating gene trees from multiple unlinked loci. Using multiple loci from different areas of the genome (e.g., mtDNA and nuDNA) can account for the different patterns of evolution experienced by each. For instance, mitochondrial alleles accumulate nucleotide substitutions several times faster than nuclear genes due to their lower effective population size, thereby completing the coalescent process much faster and becoming diagnostic of taxa more rapidly (*Sunnucks, 2000*).

Once a species tree has been inferred, additional testing is often undertaken to provide support for the proposed species' groups. A range of statistical analyses are available for testing species boundaries and, as there is currently no universally accepted way to define species, there are also a range of critiques on these methods (e.g., *Blaxter, 2004*; *Brower, 1999*; *Ebach & Holdrege, 2005*; *Lipscomb, Platnick & Wheeler, 2003*; *Seberg et al., 2003*; *Sites & Marshall, 2003*; *Sneath & Sokal, 1973*; *Tautz et al., 2002*; *Tautz et al., 2003*; *Wiens & Penkrot, 2002*; *Wiens & Servedio, 2000*; *Will, Mischler & Wheeler, 2005*; *Yang & Rannala, 2010*). Under the GLC, any evidence of lineage separation can be evidence for the existence of different species (*De Queiroz, 2007*); as such, the identification of numerous corroborating lines of evidence (through the use of multiple tests) can be seen as lending support to any species boundaries that are defined. Therefore, although no single test is currently universally accepted, the apparent need to choose a particular method is circumvented by using a selection of techniques and multiple gene regions as, under the GLC, concordance between multiple lines of evidence is seen as increasing the rigour of species delimitation.

The parastacid crayfish genera are generally highly speciose, with novel species and genetically diverse groups commonly found (e.g., *Coughran, 2005*; *Coughran et al.,*

*2012*; *Furse, Dawkins & Coughran, 2013*; *Hansen & Richardson, 2006*). The most notable exception to this is the genus *Tenuibranchiurus*, which contains the species with the smallest body size in the Parastacidae Huxley, 1879. Although it has previously been highlighted as containing genetically diverse groups (see *Dawkins et al., 2010*; *Horwitz, 1995*), this genus as currently recognised contains only the single described species *Tenuibranchiurus glypticus* (*Riek, 1951*). *Tenuibranchiurus* falls within a monophyletic clade containing the other Australian burrowing crayfish (*Gramastacus* Riek, 1972, *Geocharax* Clark, 1936, *Engaewa* Riek, 1967, *Engaeus sensu stricto* Erichson, 1846, and *Engaeus lyelli* (Clark, 1936) (distinct from other *Engaeus* species, sensu *Schultz et al., 2009*)) (*Horwitz, 1988*), and is endemic to the central-eastern coast of Australia, spanning south-east Queensland (Qld) and north-eastern New South Wales (NSW) (Fig. 1). It was first suggested by *Horwitz (1995)*, on the basis of electrophoretic and geographical differences, that previously unrecognised genetic diversity existed within the genus. Subsequently, two genetically divergent groups were identified within this region by *Dawkins et al. (2010)*, both of which showed considerable internal genetic variability. The two identified groups aligned with populations from Qld and NSW, respectively, and were suggested to represent species that diverged as a result of long-term historical geographic isolation (*Dawkins et al., 2010*). The present study seeks to quantify the genetic diversity present within *Tenuibranchiurus*, utilising molecular data across several gene regions and employing multiple species delimitation methods in order to determine the most likely species groups.

## METHODS

A total of 133 specimens were collected across 16 field localities, including the type locality for *T. glypticus*. All specimens from this study were collected under permits WITK08599510, WISP08599610, and TWB/01/2011 issued by the Department of Environment and Resource Management. DNA was extracted from specimens preserved in 70% ethanol using a variation of the cetyltrimethyl ammonium bromide/phenol-chloroform extraction protocol (*Doyle & Doyle, 1987*). Two mitochondrial gene regions (cytochrome oxidase subunit 1 (COI) and 16S rDNA (16S)) and three nuclear gene regions (glyceraldehyde-3-phosphate dehydrogenase (GAPDH), histone-3 (H3), and arginine kinase (AK)) were amplified (see Table 1 for primer list). Sequences were edited using Sequencher 4.9 (*GeneCodes, 2009*) and aligned using the MUSCLE addition in MEGA5 (*Edgar, 2004*). Alignments were then checked and edited by hand if necessary.

### Phylogenetic analyses

A total of 127 *Tenuibranchiurus* samples were sequenced for the COI gene fragment, 59 for 16S, 95 for GAPDH, 58 for H3, and 47 for AK (Table 2). Additional specimens from the genera *Gramastacus*, *Geocharax*, *Engaeus*, *Engaewa*, and *Cherax* were also sequenced for inclusion as outgroups. Where sequences from these outgroups could not be obtained (i.e., due to non-amplification), alternative sequences were retrieved from GenBank (details in Table S1). Sequences obtained in this study were deposited in GenBank under accession numbers KX669691– KX670093, KX753349.

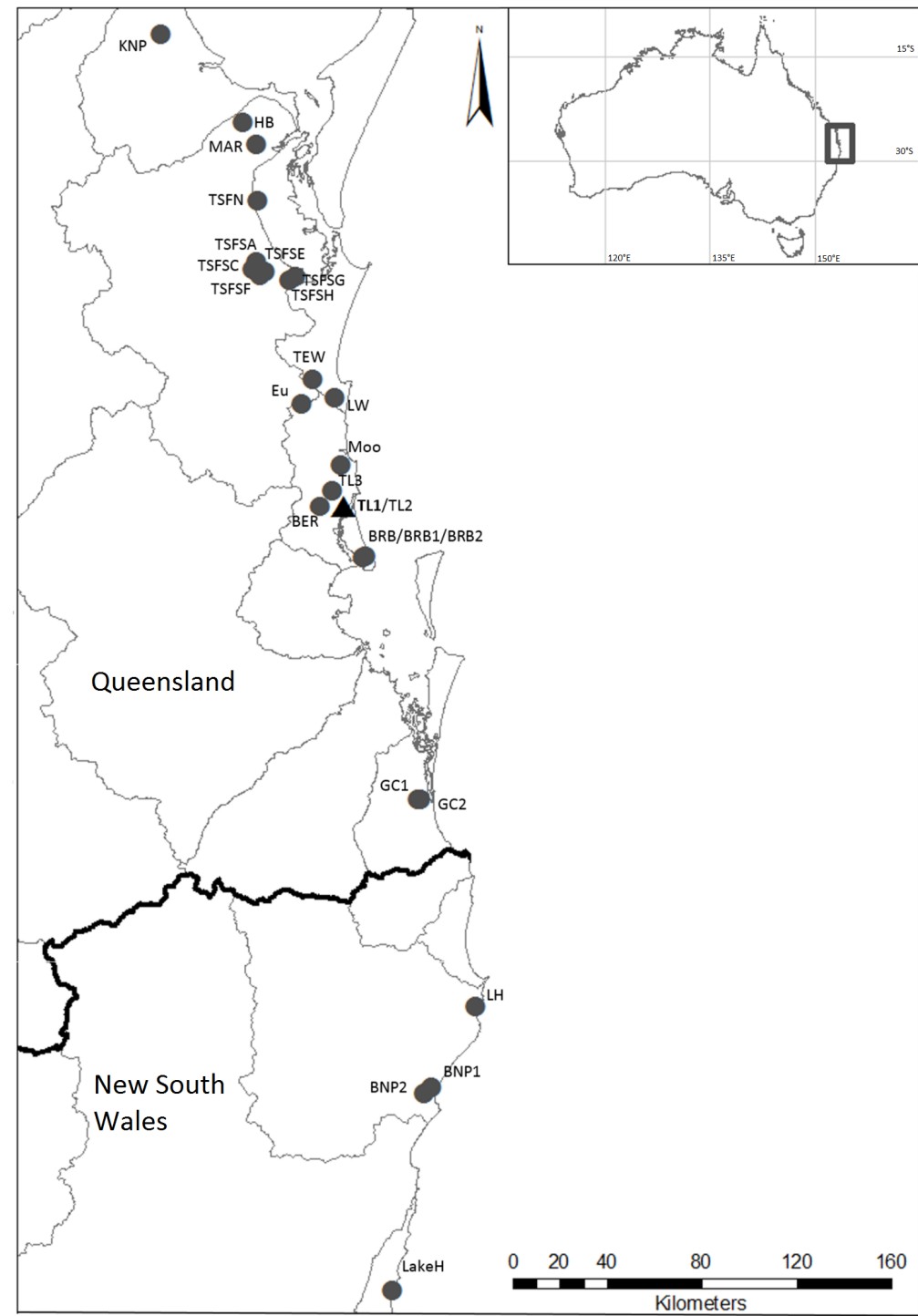

**Figure 1** **Localities where specimens of *Tenuibranchiurus* were collected during this study.** The triangle and bolded name denotes the Type Locality. Grey lines denote drainage boundaries, and the black line denotes the border between Queensland and New South Wales. Refer to Table 2 for collection details.

**Table 1** Forward and reverse primers used for PCR amplification and DNA sequencing.

| Gene region | Primers (5′ → 3′) | Reference | Fragment length |
|---|---|---|---|
| COI | CRCOI-F: CWACMAAYCATAAAGAYATTGG<br>CRCOI-R: GCRGANGTRAARTARGCTCG | *Cook, Pringle & Hughes (2008)* | 644 bp |
| 16S | 16S-ar: CGCCTGTTTATCAAAAACAT<br>16S-br: CCGGTCTGAACTCAGATCACGT | *Palumbi et al. (1991)* | 449 bp |
| GAPDH | G3PCq157-F: TGACCCCTTCATTGCTCTTGACTA<br>G3PCq981-R: ATTACACGGGTAGAATAGCCAAACTC | *Buhay et al. (2007)* | 563 bp |
| H3 | H3-AF: ATGGCTCGTACCAAGCAGACVGC<br>H3-AR: ATATCCTTRGGCATRATRGTGAC | *Colgan et al. (1998)* | 264 bp |
| AK | AKcray-F: CTACCCCTTCAACCCCTGCCTT<br>AKcray-R: CGCCCTCTGCTTCGGTGTGCTC | JW Breinholt (2012, unpublished data) | 538 bp |

### *Degree of molecular divergence*

Preliminary analyses of both individual and combined gene trees showed a prominent separation between Qld and NSW populations. In light of this, genetic distances between Qld and NSW populations, distances between these two groups and the outgroups, and distances between the outgroups were calculated using both COI and 16S data to compare the degree of separation. These distances were calculated in MEGA5 (*Tamura et al., 2011*) using the net between group mean distances with 1,000 bootstrap replicates (gamma distribution with shape parameter = 1, Maximum Composite Likelihood (MCL) model; positions containing gaps and missing data were eliminated).

## Species discovery approaches

Two types of analyses were used to obtain a best-estimate of the species-level lineages present within *Tenuibranchiurus*; namely, groupings identified through use of a concatenated alignment phylogeny (referred to henceforth as the 'combined gene tree'), and intra- versus inter-cluster variation through $\phi_{ST}$ analysis. A combined gene tree analysis was chosen over individual gene trees because, although preliminary phylogenetic analyses performed on the individual gene regions suggested that there were multiple genetic groups within *T. glypticus*, statistical support was not always strong for all genes. Therefore, in order to increase the strength of the phylogenetic signal, and thus support for branching patterns, the five gene alignments were concatenated and analysed as a single data set for phylogenetic reconstructions.

### *Combined gene tree*

Combined gene trees were inferred using both Maximum Likelihood (ML) and Bayesian analyses. Specimens were included in the data set if they were sequenced for at least four of the five genes (see Table S1). The program RAxML v. 7.4.4 through the CIPRES Science Gateway (*Miller, Pfeiffer & Schwartz, 2010*) was used to infer the ML tree, and MrBayes v. 3.2.0 (*Ronquist et al., 2012*) for the Bayesian tree. Within the ML analysis, each gene was entered as a separate DNA-partition, the GTR+CAT model used, and bootstrapping automatically halted with the majority rule criterion. For the Bayesian analysis, each gene

Table 2 Number of *Tenuibranchiurus* specimens analysed for each gene fragment from each of the sampled localities, as well as outgroup sequences included (see Table S1 for sequence details).

| State | General locality | Location ID | Number of specimens analysed | | | | |
|---|---|---|---|---|---|---|---|
| | | | COI | 16S | GAPDH | H3 | AK |
| Qld | Kinkuna National Park | KNP | – | 1 | – | – | – |
| | Hervey Bay | HB | 1 | 4 | – | 4 | 4 |
| | Maryborough | MAR | 10 | 5 | 9 | 5 | 3 |
| | Tuan State Forest (North) | TSFN | 2 | 2 | – | – | – |
| | Tuan State Forest (South) | TSFS A | 4 | 1 | 4 | 1 | 1 |
| | | C | 14 | 3 | 12 | 4 | 4 |
| | | E | 4 | 2 | 4 | 2 | 2 |
| | | F | 3 | 1 | 3 | 1 | 1 |
| | | G | 4 | – | 4 | – | – |
| | | H | 1 | – | 1 | – | – |
| | Tewantin | TEW | 7 | 3 | 7 | 4 | 4 |
| | Lake Weyba | LW | 7 | 4 | 7 | 5 | 4 |
| | Eumundi | Eu | – | 1 | – | – | – |
| | Mooloolaba | Moo | – | 1 | – | – | – |
| | Beerburrum | BER | 7 | 2 | 5 | 2 | 2 |
| | Type Locality | TL1 | – | 1 | 1 | – | – |
| | | TL2 | – | 2 | 1 | 1 | 1 |
| | | TL3 | 1 | 2 | – | – | – |
| | Bribie Island | BRB1 | – | – | 1 | – | – |
| | | BRB2 | 4 | – | – | – | – |
| | | BRB | 6 | 6 | – | 6 | 6 |
| | Gold Coast | GC1 | 8 | 3 | 5 | 5 | 3 |
| | | GC2 | 7 | 3 | 6 | 4 | 3 |
| NSW | Lennox Head | LH | 13 | 4 | 10 | 4 | 3 |
| | Broadwater National Park | BNP1 | 13 | 4 | 9 | 4 | 2 |
| | | BNP2 | 2 | 1 | 2 | 1 | – |
| | Lake Hiawatha | LakeH | 9 | 3 | 4 | 5 | 4 |
| | Total | | 127 | 59 | 95 | 58 | 47 |
| | *Gramastacus* spp. | | 6 | 10 | 4 | 7 | 4 |
| | *Geocharax* spp. | | 3 | 4 | 3 | 1 | 1 |
| | *Engaeus* spp. | | 2 | 2 | 2 | 3 | 1 |
| | *Engaewa* spp. | | 3 | 3 | 3 | 3 | 2 |
| | *Cherax* spp. | | 1 | 1 | 1 | 1 | – |
| | Total including outgroups | | 142 | 79 | 108 | 73 | 55 |

was entered as a separate partition with the Nst set as 6 for all genes except 16S (where Nst = 2), and Rates set as InvGamma for COI and GAPDH, and Gamma for 16S, H3, and AK, as determined by jModeltest v. 0.0.1 (*Posada, 2008*). Additional parameters were set as follows; two replicate Markov chain Monte Carlo analyses with four chains in each analysis (one cold, three heated), the statefreq, revmat, shape, and pinvar all unlinked, the ratepr set as variable, and the analysis set to stop when the standard deviations of the partition

frequencies <0.0099 (all effective sample size values >100, PSRF+ ≈ 1.000, and the final Ngen was 1,715,000). The same analysis was performed at least twice to verify topological convergence and homogeneity of posterior clade probabilities between runs. The first 25% of samples were discarded as burnin, with the resulting trees visualised using the program Figtree v. 1.4.0 (*Rambaut, 2012*).

### *Intra- versus inter-cluster variation*

An analysis of molecular variance (AMOVA) was used to calculate variation within and among clusters of sequences, as implemented in Arlequin v. 3.1 (*Excoffier, Laval & Schneider, 2005*). To determine what the most likely lineages were, the clades identified by the combined gene tree analysis, as well as additional splits evident within the tree that were deemed to plausibly represent lineages, were also tested, as well as groups based on the geographic division of populations (i.e., collection locality; see Table S2). The AMOVA calculates three statistics; $\phi_{ST}$, $\phi_{SC}$, and $\phi_{CT}$, all of which are based on both the haplotype frequency and genetic divergence. $\phi_{ST}$ measures variation among all populations, and $\phi_{SC}$ measures variation among populations within groups, and $\phi_{CT}$ estimates variation among groups. It has been suggested that an $F_{CT}$ value >0.95 can represent evidence for accurate species groupings (i.e., >95% of the genetic variation can be attributed to differences among groups) (*Monaghan et al., 2005*). Using the $\phi_{CT}$ estimate as a surrogate for $F_{CT}$ (as this estimate includes genetic divergence as well as haplotype frequency), this can provide an approach to delineate taxa based on population genetic analyses by interpreting the AMOVA results used to calculate intra- versus inter-cluster variation in a way analogous to $F$-statistics (*Wright, 1978*). The criterion to determine the appropriate number of lineages using this method is where an increase in the number of suggested lineages does not appreciably increase the $\phi_{CT}$ estimate for those lineages.

## Validation of lineages

In order to validate the lineages that were identified using the species discovery approaches for species-status, two methods were used; barcoding gap identification (sensu *Hebert et al., 2004*), and the $K/\theta$ method (*sensu Birky, 2013*). These two methods were chosen as they both test species boundaries by comparison of intra- and inter-lineage differences, but do so in different ways; thus allowing the results of each method to be tested and validated by the other. Only the mitochondrial data were used to validate the species hypotheses, as the nuclear gene sample sizes were limited and individually were not very informative; for instance, most of the nuclear gene trees contained numerous polytomies and thus could not be used to identify genetically divergent groups.

### *Barcoding gap*

The genetic distances between the hypothesised lineages and between specimens for both COI and 16S were calculated and visualised to determine whether a barcoding gap existed. As the intent of this method was to provide support for, or refutation of, the lineage hypotheses formed through the species discovery approaches, lineages were pre-defined based on those results and genetic distances categorised as representing either intra- or inter-lineage distances. For the purposes of this study, a barcoding gap was defined as a
clear separation (or 'gap') between the highest intra-lineage and lowest inter-lineage genetic distances measured between the suggested lineages. Although a standard threshold has been suggested by *Hebert et al. (2004)* for recognising distinct species ($10\times$ average intraspecific difference), this approach was not followed as it has been shown that there are vastly different rates of divergence for both different taxa and different genetic markers (*Avise, 2009*). Rather, a recognisable distinction between the inter- and intra-lineage distances was considered potential evidence for distinct species. Analyses were undertaken for Qld and NSW specimens separately.

Relative divergences between genetic groups were calculated in MEGA5. To determine inter-lineage divergence, the number of base substitutions per site was estimated from the net average between groups of sequences and the diversity between specimens was determined by calculating the number of base substitutions per site between each pair of sequences, both using a MCL model with 1,000 bootstrap replicates. The rate variation among sites was modelled with a gamma distribution with a shape parameter of 1, with positions containing gaps and missing data eliminated. This was performed separately for both COI and 16S, with all unique haplotypes included.

### $K/\theta$ *Method*

The species discovery hypotheses were also tested using the $K/\theta$ method (*Birky et al., 2010*; *Birky & Barraclough, 2009*; *Birky et al., 2005*). Although this method was originally developed for asexually-reproducing organisms and termed the 4X rule, it has been further developed and shown to be effective for the mtDNA region for sexually-reproducing organisms (*Birky, 2013*). This method provides a simple way of defining species groups based on specimens/populations that form clusters (i.e., clades) that are separated by genetic gaps too deep to be ascribed to random genetic drift within a species and, therefore, must be due to diversifying selection or long-term physical isolation (*Apte, Smith & Wallis, 2007*).

Using the groups from the species discovery hypotheses, sister clades were identified and statistical support for these was tested. Sequence divergences were estimated within ($d$) and between each sister clade using uncorrected p-distances calculated in MEGA5. Nucleotide diversity ($\pi$) was then calculated using $\pi = dn/(n-1)$, where n is the number of samples per clade. Theta ($\theta$) was then estimated as $\theta = 2Ne\mu$ (where $Ne$ is the effective populations size and $\mu$ is mutation rate per base pair per generation) by calculating $\pi/(1 - 4\pi/3)$ within each clade. If $d = 0$ (as it did for one clade in this study), then $\pi$ can alternatively be calculated as $2/Ln(n-1)$, where $L$ is the length of the sequence. $K$ was then calculated for each sister-clade comparison (using MEGA5) as the uncorrected net between group mean distance, with this divided by the highest $\theta$ in the comparison (as this is the more conservative approach) to provide $K/\theta$. Where sister clades were poorly defined in the tree, $K$ was estimated between all potential sister clades in the polytomy, with the clade of the lowest $K$ considered to be the sister clade. Finally, following the method of *Birky (2013)*, if the $K/\theta$ value was greater than 4, then the sister clades were accepted as different lineages.

**Table 3** Estimates of net evolutionary divergence between groups of COI (below diagonal) and 16S (above diagonal) sequences with a MCL model.

|  | Qld | NSW | *Geocharax* | *Gramastacus* | *Engaeus* | *Engaewa* | *Cherax* |
|---|---|---|---|---|---|---|---|
| Qld | – | 0.127 | 0.14 | 0.161 | 0.101 | 0.175 | 0.24 |
| NSW | 0.160 | – | 0.113 | 0.117 | 0.072 | 0.191 | 0.24 |
| *Geocharax* | 0.156 | 0.164 | – | 0.129 | 0.067 | 0.212 | 0.257 |
| *Gramastacus* | 0.185 | 0.206 | 0.203 | – | 0.081 | 0.244 | 0.242 |
| *Engaeus* | 0.109 | 0.086 | 0.137 | 0.117 | – | 0.138 | 0.189 |
| *Engaewa* | 0.164 | 0.154 | 0.160 | 0.169 | 0.103 | – | 0.347 |
| *Cherax* | 0.256 | 0.256 | 0.261 | 0.294 | 0.195 | 0.228 | – |

## RESULTS

### Degree of molecular divergence

The genetic distances calculated between the Qld and NSW groups using COI and 16S were 16.0% and 12.7%, respectively (Table 3). These distances were as large as, or in some cases larger than, the distances calculated between these two groups and other closely related genera. Furthermore, some distances between pairs of the other genera were smaller than those between the Qld and NSW groups for both COI and 16S (e.g., *Geocharax* versus *Engaeus* = 13.7% and 6.7%, *Gramastacus* versus *Engaeus* = 11.7% and 8.1%; Table 3).

### Species discovery

Groups that are identified as potentially representing distinct species will be referred to herein as Lineages, and will form the groups to be analysed through lineage validation methods.

#### *Combined gene tree*

Although not all groupings were statistically supported, both the ML and Bayesian combined gene trees suggested the presence of multiple groups within Qld and NSW, and displayed the same topologies (Fig. 2). Six clades were evident within the Qld populations, with the monophyly of all but two highly supported (as these were represented by single specimens). The first clade included Maryborough and some Tuan State Forest specimens (Lineage 1; BS 90%, Pp 1), and the second contained the remaining Tuan State Forest specimens as well as Bribie Island, Type Locality, and some Beerburrum specimens (Lineage 2; BS 96%, Pp 1). The two groups for which monophyly could not be established were represented by the remaining Beerburrum specimens (Lineage 3) and Hervey Bay (Lineage 4). The final two clades consisted of Tewantin and Lake Weyba specimens (Lineage 5; BS 100%, Pp 1) and Gold Coast specimens (Lineage 6; BS 100%, Pp 1). There was also some geographic structuring evident within each of the clades.

The two monophyletic clades evident within the NSW populations were strongly supported, and form Lineage 7 (Lennox Head) and Lineage 8 (Lake Hiawatha, Broadwater National Park 1 & 2) (Fig. 2). Although there was some structuring evident within Lineage 8, the branching patterns were very shallow and were therefore not explored as potential distinct lineages.

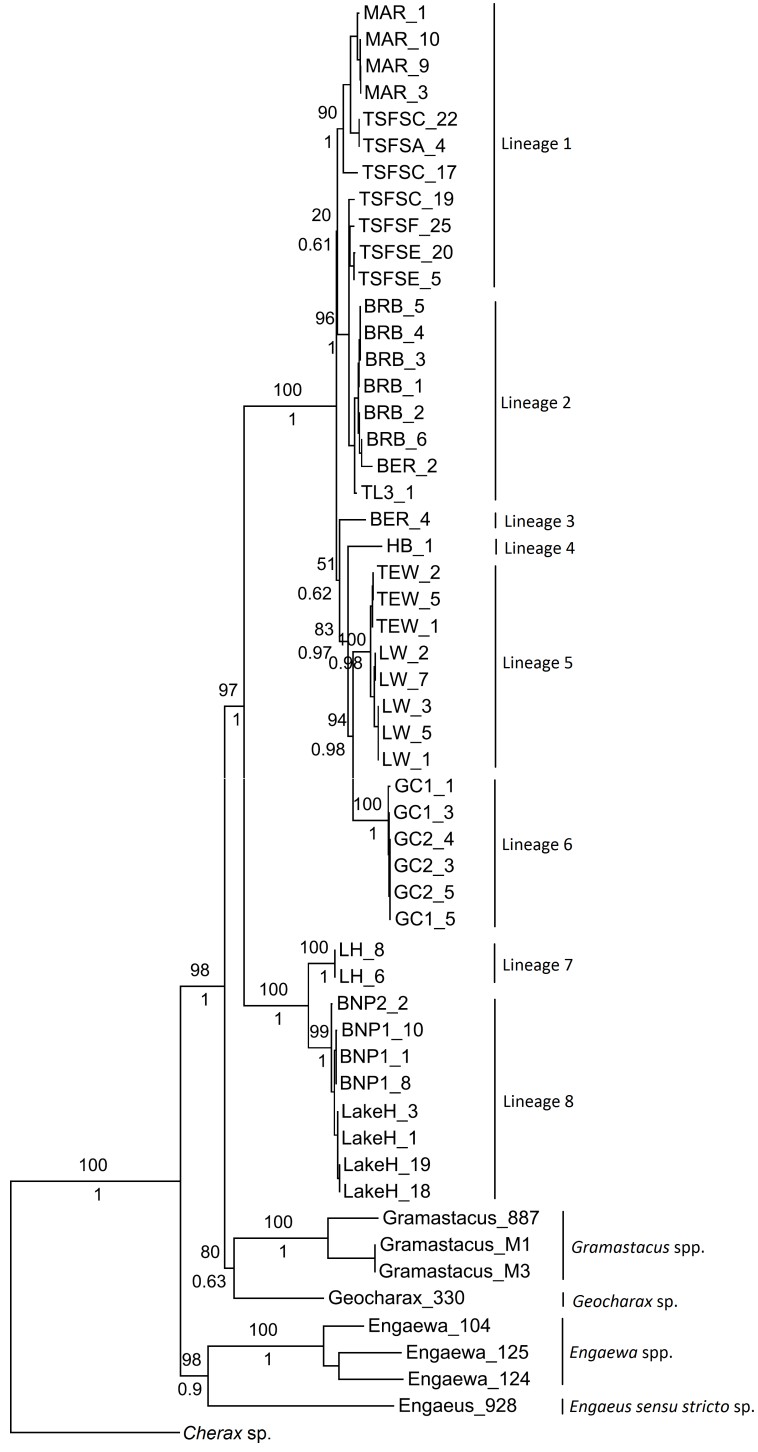

**Figure 2 Maximum likelihood phylogeny showing the proposed lineages for Queensland (Lineages 1 through 6) and New South Wales (Lineages 7 and 8).** Bootstrap values from the maximum likelihood analysis are shown above the branches, and posterior probabilities from the Bayesian analysis are shown below branches for the major nodes, as both analyses produced identical topologies for the major nodes.

**Table 4 Summary of possible lineages based on $\phi$-statistics for Qld specimens using COI and 16S data.**
See Table S2 for explanation of how potential lineages were determined. Where specimens from the same collection locality are split into two or more groups, details are included below the table for clarification.

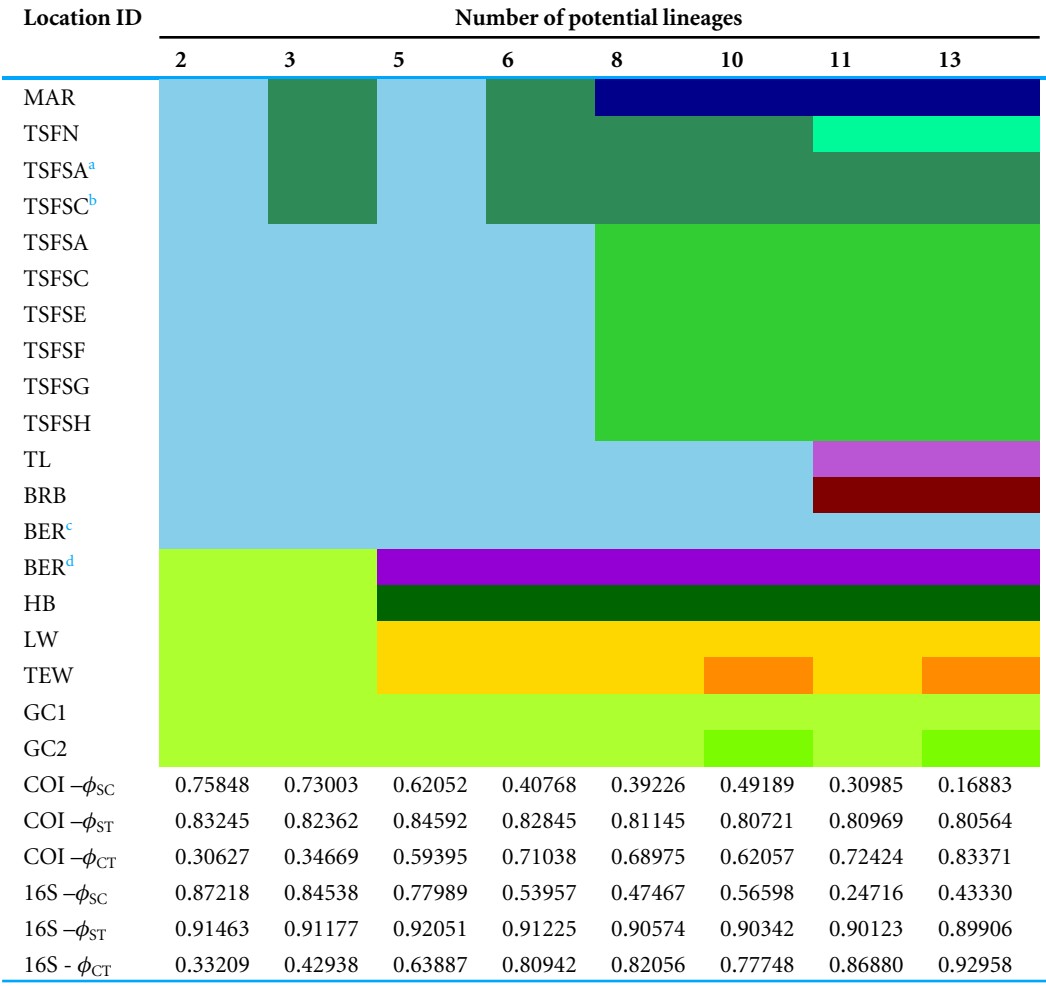

| Location ID | Number of potential lineages | | | | | | | |
|---|---|---|---|---|---|---|---|---|
| | 2 | 3 | 5 | 6 | 8 | 10 | 11 | 13 |
| MAR | | | | | | | | |
| TSFN | | | | | | | | |
| TSFSA[a] | | | | | | | | |
| TSFSC[b] | | | | | | | | |
| TSFSA | | | | | | | | |
| TSFSC | | | | | | | | |
| TSFSE | | | | | | | | |
| TSFSF | | | | | | | | |
| TSFSG | | | | | | | | |
| TSFSH | | | | | | | | |
| TL | | | | | | | | |
| BRB | | | | | | | | |
| BER[c] | | | | | | | | |
| BER[d] | | | | | | | | |
| HB | | | | | | | | |
| LW | | | | | | | | |
| TEW | | | | | | | | |
| GC1 | | | | | | | | |
| GC2 | | | | | | | | |
| COI $-\phi_{SC}$ | 0.75848 | 0.73003 | 0.62052 | 0.40768 | 0.39226 | 0.49189 | 0.30985 | 0.16883 |
| COI $-\phi_{ST}$ | 0.83245 | 0.82362 | 0.84592 | 0.82845 | 0.81145 | 0.80721 | 0.80969 | 0.80564 |
| COI $-\phi_{CT}$ | 0.30627 | 0.34669 | 0.59395 | 0.71038 | 0.68975 | 0.62057 | 0.72424 | 0.83371 |
| 16S $-\phi_{SC}$ | 0.87218 | 0.84538 | 0.77989 | 0.53957 | 0.47467 | 0.56598 | 0.24716 | 0.43330 |
| 16S $-\phi_{ST}$ | 0.91463 | 0.91177 | 0.92051 | 0.91225 | 0.90574 | 0.90342 | 0.90123 | 0.89906 |
| 16S - $\phi_{CT}$ | 0.33209 | 0.42938 | 0.63887 | 0.80942 | 0.82056 | 0.77748 | 0.86880 | 0.92958 |

**Notes.**
[a] TSFSA;4.
[b] TSFSC;8,17,22.
[c] BER;1,2,5.
[d] BER;3,4,6,7.

### Intra- versus inter-cluster variation

A total of eight lineage arrangements was deemed plausible based on apparent genetic groupings and collection localities, and were tested using AMOVAs (Table 4). The process of assigning the potential lineages is outlined in Table S2, where a hierarchical approach was taken to split the tree into major genetic groups, minor genetic groups, and geographic localities. As there was no logical reason for combining the NSW lineages for the AMOVA analysis based on either the phylogenetic or geographic information, the NSW populations were considered to consist of the LH lineage and the LakeH/BNP lineage. Further testing, however, was considered appropriate to determine the lineages present within Qld. Figure 3 shows an increase in the $\phi_{CT}$ estimate, with a plateau reached at six lineages for both the COI

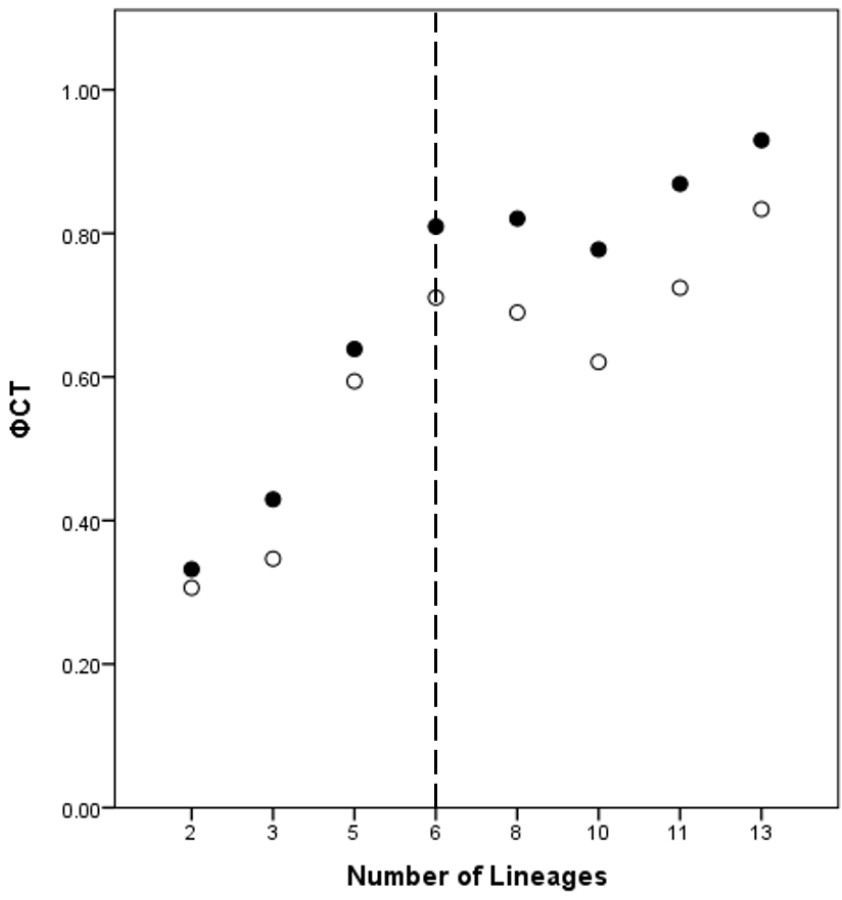

**Figure 3** $\phi_{CT}$ **values for potential lineages for both COI (open circles) and 16S (black circles) for Queensland specimens.** The dotted line indicates the most likely delimitation at six lineages.

and 16S estimates. These six Qld lineages represent the most parsimonious arrangement of the specimens into lineages.

### Species discovery hypothesis

As the combined gene tree was inferred using only specimens that were successfully sequenced for at least four of the five genes, not all collection localities were represented on the tree (i.e., TSFN, KNP, Moo, Eu). Of these localities, only TSFN was represented in the AMOVA analysis, as the remaining localities were represented by a single sequence and therefore could not be included in the AMOVA. In order to assign these populations to a lineage for further testing, the individual gene trees and haplotype networks were examined and the localities were designated through the closest phylogenetic connection (data not shown). Both of the species discovery approaches suggested the presence of eight lineages (six in Qld and two in NSW; Table 5), and formed the lineages to be validated.

**Table 5  Lineages assigned through two species discovery approaches and the final lineage hypothesis, for Queensland and New South Wales localities.** Dashes indicate where a population was not included.

| Location ID | Combined gene tree | AMOVA | Lineage hypothesis |
|---|---|---|---|
| KNP | – | – | |
| TSFN | – | | Lineage 1 |
| MAR | | Lineage 1 | |
| TSFS | Lineage 1 | | |
| Moo | – | – | |
| TSFS | | | |
| TL | | | Lineage 2 |
| BRB | Lineage 2 | Lineage 2 | |
| BER | | | |
| BER | Lineage 3 | Lineage 3 | Lineage 3 |
| HB | Lineage 4 | Lineage 4 | Lineage 4 |
| TEW | | Lineage 5 | |
| LW | Lineage 5 | | Lineage 5 |
| Eu | – | – | |
| GC | Lineage 6 | Lineage 6 | Lineage 6 |
| LH | Lineage 7 | Lineage 7 | Lineage 7 |
| BNP | | | |
| LakeH | Lineage 8 | Lineage 8 | Lineage 8 |

## Validation of Lineages

### Barcoding gap

The COI data showed some overlap of the intra- and inter-lineage estimates within Qld, resulting in no usable barcoding gap for lineage separation (Fig. 4A). Where the overlap occurred, the low inter-lineage estimates were attributable to the Lineage 1 vs. Lineage 2 comparison, and the high intra-lineage estimates were seen between specimens within Lineage 1. However, many estimates between these two lineages fell in the higher range of the inter-lineage estimates as well as the low range.

The 16S data for Qld populations showed a clearer relationship between lineages (Fig. 4C). Although there was a very small overlap between the intra- and inter-lineage distances (occurring between two specimens from Lineage 1), this represented an overlap of less than 0.01%. When the existence of this overlap was disregarded, there was a small gap at 2.8–3.0%. However, despite there not being a distinguishable gap due to the overlap, identification of the majority of lineages through the comparison of intra- and inter-lineage distances was clear and distinguishable.

When the estimates within and between Lineage 1 and 2 specimens were removed from both the COI and 16S data (with the comparison between these two lineages and all other lineages remaining), a clear barcoding gap was seen in both data sets (Fig. 4B, Fig. 4D). For COI, the gap occurred between 1.7–4.7%, and between 0.9–3.5% for 16S. This shows that all other Qld groups (i.e., Lineage 3 through 6) represent clear lineages based on the barcoding approach using both COI and 16S data.
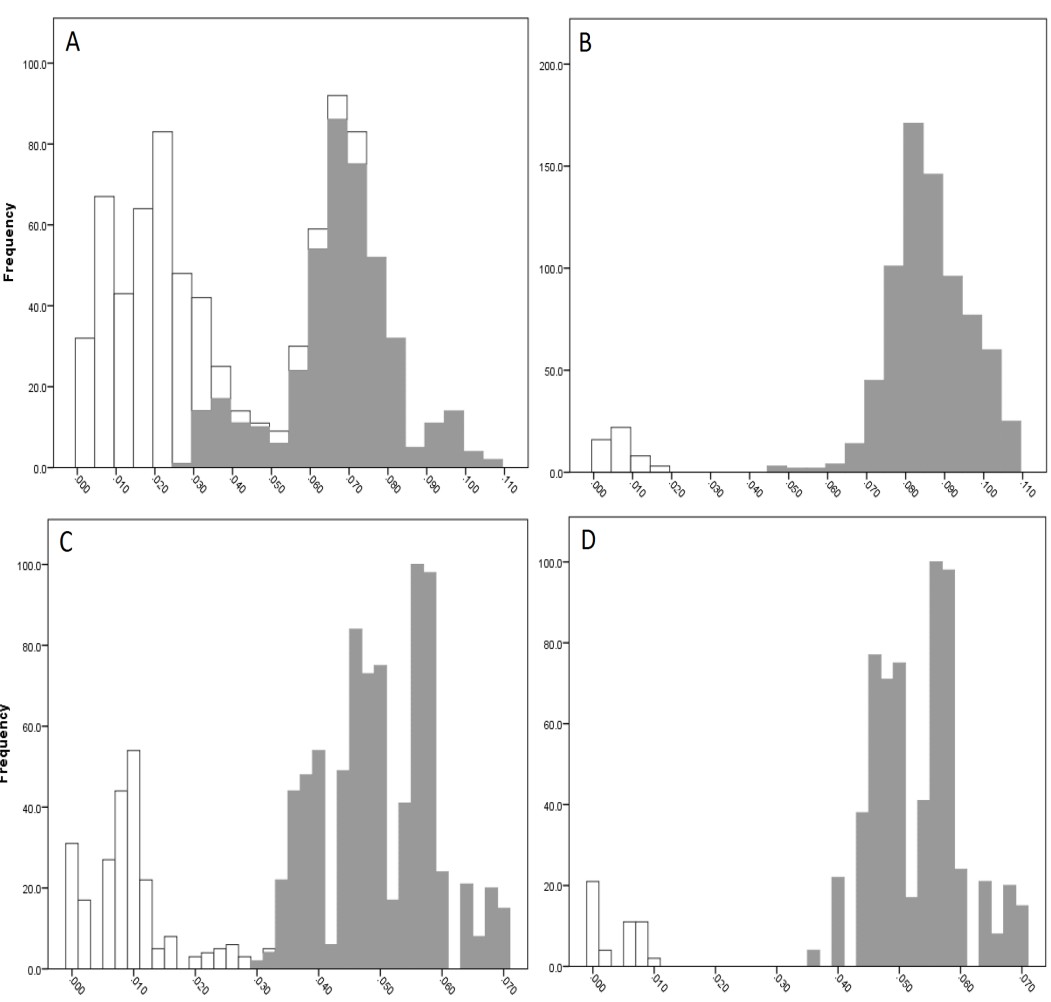

**Figure 4** **Intra- and inter-lineage genetic distance estimates (white and grey, respectively) for Queensland lineages showing (A) COI estimates for all lineages, (B) COI estimates without comparisons between Lineage 1 and 2, (C) 16S estimates for all lineages, and (D) 16S estimates without comparisons between Lineage 1 and 2.**

For NSW populations, there was a clear barcoding gap between the two lineages (i.e., Lineage 7 and 8), occurring between 1.5 and 6.6% for the COI data and 0.7–3.0% for the 16S data (Fig. 5).

### $K/\theta$ method

The sister clades within Qld and NSW were tested using the $K/\theta$ method for a delimitation of eight lineages (six from Qld, two from NSW) using both COI and 16S data (Table 6). In some instances, sister clades that were defined by the lowest $K$-distance (as they were ambiguous based on the combined gene tree) differed between the COI and 16S datasets. In these cases, only the relevant $K/\theta$ comparison for the applicable gene was calculated.

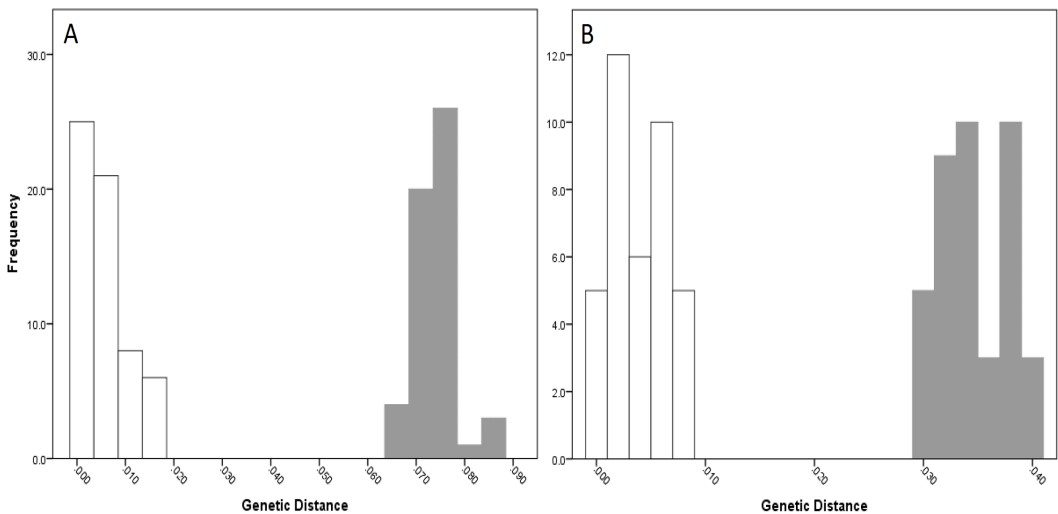

**Figure 5** Intra- and inter-lineage genetic distance estimates (white and grey, respectively) for New South Wales lineages showing (A) COI and (B) 16S estimates for all lineages.

**Table 6** $K/\theta$ values for both COI and 16S for comparisons between sister clades within Queensland and New South Wales. Where specimens from the same collection locality are split into two or more lineages, details are included below the table for clarification. Dashes are used where sister clades differ between COI and 16S.

| Sister Clade 1 | Sister Clade 2 | $K/\theta$ | |
| --- | --- | --- | --- |
| | | **COI** | **16S** |
| Lineage 1 | Lineage 2 | 0.78 | 1.41 |
| Lineage 2 | Lineage 1 | 0.78 | 1.41 |
| Lineage 3 | Lineage 1 | – | 1.67 |
| | Lineage 5 | 6.99 | – |
| Lineage 4 | Lineage 5 | 7.18 | – |
| | Lineage 6 | – | 32.84 |
| Lineage 5 | Lineage 6 | 6.71 | – |
| | Lineage 2 | – | 4.92 |
| Lineage 6 | Lineage 5 | 6.71 | 8.24 |
| Lineage 7 | Lineage 8 | 16.03 | 6.48 |
| Lineage 8 | Lineage 7 | 16.03 | 6.48 |

**Notes.**

Lineage 1 = MAR&TSFN&TSFSA (specimen 4) &TSFSC (specimens 8,17,22).

Lineage 2 = TSFSA-H (specimens 1-3,5-7,9-12,14,16,18-21,23-30) & BRB & TL & BER (specimens 1,2,5).

Lineage 3 = BER (specimens 3,4,6,7).

### Lineage assignment

Although there was some ambiguity in the barcoding analysis of the Qld COI data regarding the separation of Lineage 1 and 2, the 16S data showed support for the species discovery lineage hypothesis. Because of the deeper phylogenetic inferences provided by 16S in addition to the fact that there were many genetic distances within and between Lineage 1 and 2 falling within the expected distributions, the lineage hypothesis for Qld populations was considered supported by this analysis (Table 7). The two NSW lineages were clearly

**Table 7** **The species discovery lineage hypothesis and two lineage validation methods, with the final assignment of lineages for Queensland and New South Wales localities.** Dashes indicate where a population was not included.

| Location ID | Lineage hypothesis | Barcoding gap | $K/\theta$ | Final lineage assignment |
|---|---|---|---|---|
| KNP | | | – | |
| TSFN | Lineage 1 | Lineage 1 | | Lineage 1 |
| MAR | | | Lineage 1/2 | |
| TSFS | | | | |
| Moo | | | – | |
| TSFS | | | – | |
| TL | Lineage 2 | Lineage 2 | | Lineage 2 |
| BRB | | | Lineage 1/2 | |
| BER | | | | |
| BER | Lineage 3 | Lineage 3 | Lineage 3 | Lineage 3 |
| HB | Lineage 4 | Lineage 4 | Lineage 4 | Lineage 4 |
| TEW | | | Lineage 5 | |
| LW | Lineage 5 | Lineage 5 | | Lineage 5 |
| Eu | | | – | |
| GC | Lineage 6 | Lineage 6 | Lineage 6 | Lineage 6 |
| LH | Lineage 7 | Lineage 7 | Lineage 7 | Lineage 7 |
| BNP | Lineage 8 | Lineage 8 | Lineage 8 | Lineage 8 |
| LakeH | | | | |

separate based on both the COI and 16S data and were therefore also supported (Table 7). In the $K/\theta$ analysis, all lineages were supported by both genes with the exception of the split between Lineage 1 and 2 (both genes), and Lineage 1 and 3 (16S) (Table 7).

## DISCUSSION

### Phylogenetic relationships

Based on a preliminary data set, *Dawkins et al. (2010)* highlighted the presence of two genetically divergent groups within *Tenuibranchiurus* and from this suggested the potential presence of two distinct species within the genus. The phylogenetic reconstruction of this study supports the presence of these two divergent groups; however, the larger data set used, as well as the additional nuclear genes analysed, suggests that the recognition of the two groups should be at a generic, rather than specific, level. Inclusion of the most closely-related genera (i.e., *Gramastacus*, *Geocharax*, *Engaeus*, and *Engaewa*) in the analyses shows that the genetically divergent entities represented by the Qld and NSW groups each form monophyletic clades, to the exclusion of all other genera. While the splitting of a monophyletic grouping into two genera is arguably arbitrary, the degree of divergence suggests it is warranted in this case. The only other studies to suggest a novel parastacid genus based on molecular analyses were undertaken by *Schultz et al. (2007)* and *Hansen & Richardson (2006)*. The study of *Schultz et al. (2007)* proposed a potential new genus (which was simply referred to as *E. lyelli*, as *Engaeus lyelli* was the only species in the

divergent lineage) based on it being a highly divergent lineage within their phylogeny and found support for this through comparisons of genetic distances, in a similar fashion to this study. However, the validity of this genus was never thoroughly tested, nor were the authors confident that complete taxon sampling had been achieved. *Hansen & Richardson (2006)* included molecular data in their rationale for erecting two new genera (*Spinastacoides* and *Ombrastacoides*), but simply stated that their criteria for recognition of a genus was "a substantial degree of genetic difference, either in the 16S data or in the allozyme data", combined with support from their morphological data (*Hansen & Richardson, 2006*, pg 719). As such, due to the widespread and extensive collecting undertaken and the application of methods to identify and test lineage hypotheses employed in this study, it is the first to propose and validate the presence of a novel parastacid genus using a systematic and thorough testing of molecular data.

Although it is difficult to define what degree of separation is necessary between genera at a molecular level (*Rach et al., 2008*), based on the genetic distances presented in this study there is strong support for division at genus level. For instance, the genetic distance between Qld and NSW populations is larger than that seen between *Engaeus* and both *Geocharax* and *Gramastacus* for the COI and 16S gene fragments, and between *Engaewa* and both *Geocharax* and *Engaeus* for COI. Other genera also show smaller genetic distances when compared to either Qld or NSW than these two groups do with each other. Regardless of which genera were genetically closer to each other, the distance between Qld and NSW is at least as large as those between existing genera (Table 3), thereby supporting their separation into two distinct genera.

## Species identification

Both of the species discovery approaches established the presence of the same eight lineages across Qld and NSW specimens of *Tenuibranchiurus*. Of these, Lineages 3 through 8 were highly supported by the two lineage validation methods used. However, support for the distinction between Lineages 1 and 2 was dependent upon the method and gene used. Using the barcoding approach, it has been found that recently-diverged species are harder to distinguish than older species (*Lou & Golding, 2010*; *Meyer & Paulay, 2005*; *Yassin et al., 2010*), with problems most likely attributable to incomplete lineage-sorting resulting in the lack of a barcoding gap (*Van Velzen et al., 2012*). Additionally, when using both the barcoding and $K/\theta$ methods, the high levels of genetic diversity found within each lineage (rather than low levels between them) may have resulted in these two lineages not being strongly supported. Alternatively, retained ancestral variation between two recently-diverged clades may mask their current genetic isolation using the $K/\theta$ method, as divergence will follow a continuum and therefore no single percentage will work in every case (*Druzhinina et al., 2012*). Although this method has proven useful for other studies of sexually-reproducing organisms (e.g., *Leasi et al., 2013*; *Marrone, Lo Brutto & Arculeo, 2010*; *Reniers et al., 2013*), the results presented here suggest that it may not be suitable for delineating between some species where intraspecific diversity is high. In light of this, and considering the support shown by the species discovery lineages suggested and the

**Table 8** **Existing and putative species identified within *Tenuibranchiurus* and the newly proposed *Gen. nov.*.**

| Species | Location |
| --- | --- |
| *Tenuibranchiurus* sp. nov. 1 | Kinkuna National Park |
| | Tuan State Forest North |
| | Maryborough |
| | Tuan State Forest South |
| | Mooloolaba |
| *Tenuibranchiurus glypticus* | Tuan State Forest South |
| | Type Locality |
| | Bribie Island |
| | Beerburrum |
| *Tenuibranchiurus* sp. nov. 2 | Beerburrum |
| *Tenuibranchiurus* sp. nov. 3 | Hervey Bay |
| *Tenuibranchiurus* sp. nov. 4 | Tewantin |
| | Lake Weyba |
| | Eumundi |
| *Tenuibranchiurus* sp. nov. 5 | Gold Coast |
| *Gen. nov.* sp. nov. 1 | Lennox Head |
| *Gen. nov.* sp. nov. 1 | Broadwater National Park |
| | Lake Hiawatha |

barcoding results, Lineage 1 and 2 are accepted as independently evolving lineages and, therefore, putative species.

Based on our results, the genus *Tenuibranchiurus* is represented only by specimens collected from Queensland. As such, *Tenuibranchiurus glypticus* remains a valid species and it is represented by populations grouped with samples from the Type Locality. Five new putative species were identified within *Tenuibranchiurus* (Table 8). Specimens collected from New South Wales belong to a putative newly-proposed genus with two new putative species (Table 8). Until a formal description is completed, the new genus will be referred to as *Gen. nov.*

## CONCLUSIONS

Although genetic diversity within *Tenuibranchiurus* has previously been reported, no quantification of this diversity had been undertaken. The multi-gene approach taken by this study and use of several different analytical methods has identified not only several putative species within the formerly monotypic *Tenuibranchiurus*, but a new genus with two putative species. Although species identification of freshwater crayfish has traditionally been made through morphological methods, the use of molecular methods in this study allowed the potential pitfalls of plastic and/or cryptic morphological forms within crayfish to be avoided, and will contribute in the development of a standardised method for dealing with species identification within other freshwater crayfish.

## ACKNOWLEDGEMENTS

Many thanks are given to the volunteers that helped with the field work; Dr Seanan Wild, Amanda Dawson, Dr Dianna Virkki, and Shane Howard. We are also grateful for additional genetic material provided by Dr Andrew Bentley and Dr Quinton Burnham.

### Funding

This project was completed as part of a PhD project undertaken by the principal author, funded by an Australian Postgraduate Award (2009) and the Griffith School of Environment, Gold Coast campus, Queensland, Australia. There was no additional external funding received for this study. The funders had no role in study design, data collection and analysis, decision to publish, or preparation of the manuscript.

### Grant Disclosures

The following grant information was disclosed by the authors:
Australian Postgraduate Award (2009).
Griffith School of Environment, Gold Coast campus, Queensland, Australia.

### Competing Interests

Jane M. Hughes is an Academic Editor for PeerJ.

### Author Contributions

- Kathryn L. Dawkins conceived and designed the experiments, performed the experiments, analyzed the data, wrote the paper, prepared figures and/or tables, reviewed drafts of the paper.
- James M. Furse conceived and designed the experiments, reviewed drafts of the paper.
- Clyde H. Wild reviewed drafts of the paper.
- Jane M. Hughes conceived and designed the experiments, contributed reagents/materials/analysis tools, reviewed drafts of the paper.

### Field Study Permissions

The following information was supplied relating to field study approvals (i.e., approving body and any reference numbers):

All specimens from this study were collected under permits WITK08599510, WISP08599610, and TWB/01/2011 issue by the Department of Environment and Resource Management.

### DNA Deposition

The following information was supplied regarding the deposition of DNA sequences:

Sequences obtained in this study were deposited in GenBank under accession numbers KX669691– KX670093, KX753349.

## Data Availability

The raw data has been supplied as a Supplementary File.

## Supplemental Information

Supplemental information for this article can be found online at http://dx.doi.org/10.7717/peerj.3310#supplemental-information.

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
