# Peer review of "A novel genus and cryptic species harboured within the monotypic freshwater crayfish genus Tenuibranchiurus Riek, 1951 (Decapoda: Parastacidae)"

_PeerJ, doi:10.7717/peerj.3310_

## Round 0.1 · original submission · Minor Revisions

I have now received three thorough reviews of your paper in addition to reading it over myself. I am happy to report we all think this an interesting and well-done study. The sampling is impressive, the analyses well done, and the study targets a much needed area of effort for crayfish biodiversity and phylogenetic work. Note there are, however, a number of issues that need to be dealt with before acceptance. The central issue, in my view, is that raised by reviewer 1 with respect to a more stringent description/articulation of what you mean by 'species delimitation' as this is the core focus of your paper. I have no doubt you can efficiently deal with this and the other issues raised by reviewers and I look forward to your revised draft. Please also note reviewer 2 has an annotated manuscript that you will need to download to see the full extent of those suggestions.

Good luck with your revision.

·

Basic reporting

Dawkins et al. take a molecular phylogenetic approach to describe the genetic diversity within Tenuibranchiurus and find evidence for multiple divergent lineages and potentially two independent genera. The manuscript is very nicely written and the background information and literature cited is appropriate. The methods and analyses are sound and the authors take care not to overstate the implications of their results. However the authors should consider describing the species identified as something like ‘potential’ or ‘putative’ species rather than ‘species’ in the Discussion and Conclusions sections, until other lines of evidence corroborate.

Experimental design

This study extends the author’s findings in Dawkins et al. 2010 by including additional sampling of localities and genes as well as several tests of lineage delimitation. It is unclear to me if the data from Dawkins et al. 2010 were included in this analysis.
The authors should consider clarifying the terms and approaches for ‘species delimitation’ and ‘testing of lineages’ in the Methods section. It seems ‘species delimitation’ is being used here to mean lineage assignment or identification of genetic structure and ‘testing of lineages’ to mean ‘species delimitation’ in the sense of testing the hypotheses that these lineages are independent, genetically distinct, etc. The authors could consider the terms ‘discovery’ and ‘validation’ as in Carstens et al. (Molecular Ecology (2013) 22, 4369–4383). Also, given the array of molecular species delimitation methods available (and discussed in the Introduction) that could be applied under the General Lineage Concept, perhaps the authors should provide practical or theoretical justification for choosing these two.

Validity of the findings

no comment

Additional comments

Additional suggestions:

Line 61 – ‘Ne’ could be changed to ‘effective population size’
Line 81 - Perhaps this could be clarified that this is ‘smallest’ is in terms of physical size, since the rest of the paragraph is about clade size
Line 110 – Should these primers sequences be made available here?
Line 141- I am wondering if the use of the term ‘combined gene tree’ invokes the idea of summarizing individual gene trees in some way to estimate a ‘species tree’. Perhaps a more appropriate term would be something like ‘concatenated alignment phylogeny.’ In that sense, Line 147 could be changed to ‘gene alignments were contatenated.’
Line 155 – Bootstrapping automatically halted at what point? How many bootstrap replicates were generated? What substitution models were used in the Bayesian analysis?
Line 163 – Was the posterior summarized as an MCC or consensus tree? Or were clade frequencies mapped onto the ML tree?
Line 164 – Were both ML and Bayesian trees used for downstream analysis/ lineage assignment?
Line 172- Table S2 can be referenced here
Line 174 – Missing the word ‘variation’ in ‘Phi-SC measures among’
Line 181 – Is there a function to perform this analysis in Arlequin? What criteria were used to make the cutoff?
Line 191 – The authors could consider including the individual gene trees as Supplementary Figures, and in either case they could describe how those gene trees were estimated.
Line 212 – Were these 1000 bootstrap replicates?
Line 214 - It could be made clear that this was performed for 16S and COI separately
Line 238 - Is the cutoff of 4 here following recommendations from another paper?
Line 387- I assume ‘both gene fragments’ here mean 16S and COI, but that should be made clear
Lines 389-390- Table 2 can be referenced here
Line 435- ‘ …addition genus…’ should read ‘additional’
Figure 2 – Is this the maximum-likelihood phylogeny or a consensus tree from the Bayesian analysis?

·

Basic reporting

No comments

Experimental design

No comments

Validity of the findings

Discussion and Conclusion should be improved because they are a little bit repetitive.

Additional comments

This paper present an important contribution to the crayfish taxonomy, since the authors proposed the establishment of a novel genus with two new species and five new species for the genus Tenuibranchiurus. Methods are clear and the data analysis was well employed with robust and reliable results. I pointed some modifications in the PDF file and I highly encourage the publication of this paper after the small modifications.

·

Basic reporting

This article is well written, comprehensively referenced with good structure and well designed tables and figures. The raw data is included and the methods, which are complex, are detailed and carefully explained.

Experimental design

This research is an important contribution to the phylogeny and taxonomy of freshwater crayfish. A monotypic genus has been carefully sampled, and five gene regions were analysed to identify a total of two genera containing eight species in this cryptic species complex.

The methods were rigorous and well explained, showing all possible hypotheses (in Table S2) and using a variety of analytical approaches (lineage testing, barcoding gap identification and K/θ) that converge on the eventual solution which is robust and convincing.

Validity of the findings

The findings appear valid and I commend the authors on the collection of 127 burrowing crayfish, which is not an easy task. The subsequent analysis, both genetic and statistical is comprehensive and thoroughly convincing.

A major revision of the genus is sure to follow and I look forward to further publications on this unique and now even more interesting taxa.

Additional comments

Minor typos and a suggestion:

Line 103 : issued instead of issue?
Line 160: sentence doesn't flow, seems to be missing a right handed parenthesis = )
Figure 1: given the emphasis on the separation of QLD and NSW samples, it may be helpful to add the state boundary to the map, especially for the international audience. This is just a suggestion as it is possible to figure out where it is from the lineage information.

---

## Round 0.2 · accepted · Accept

Thank you for your careful revision of your manuscript. You've done a great job of incorporating the feedback from the previous round of reviews. I feel your paper is now, therefore, ready for publication.